# COOBoostR: An Extreme Gradient Boosting-Based Tool for Robust Tissue or Cell-of-Origin Prediction of Tumors

**DOI:** 10.3390/life13010071

**Published:** 2022-12-27

**Authors:** Sungmin Yang, Kyungsik Ha, Woojeung Song, Masashi Fujita, Kirsten Kübler, Paz Polak, Eiso Hiyama, Hidewaki Nakagawa, Hong-Gee Kim, Hwajin Lee

**Affiliations:** 1Biomedical Knowledge Engineering Laboratory, Seoul National University, Seoul 08826, Republic of Korea; 2Dental Research Institute, Seoul National University, Seoul 08826, Republic of Korea; 3Department of Medicine, Hanyang University, Seoul 04763, Republic of Korea; 4Laboratory of Cancer Genomics, RIKEN Center for Integrated Medical Sciences, Yokohama 230-0045, Japan; 5Berlin Institute of Health at Charité—Universitätsmedizin Berlin, Charitéplatz 1, 10117 Berlin, Germany; 6Department of Hematology, Oncology and Cancer Immunology, Charité—Universitätsmedizin Berlin, Hindenburgdamm 30, 12203 Berlin, Germany; 7Broad Institute of MIT and Harvard, Cambridge, MA 02142, USA; 8Center for Cancer Research, Massachusetts General Hospital, Charlestown, MA 02129, USA; 9Department of Medicine, Harvard Medical School, Boston, MA 02115, USA; 10Department of Oncological Sciences, Icahn School of Medicine at Mount Sinai, 1425 Madison Ave., New York, NY 10029, USA; 11Natural Science Center for Basic Research and Development, Hiroshima University, Hiroshima 739-8529, Japan

**Keywords:** biocomputational method, bioinformatics-based prediction of cell-of-origin, genomics, epigenomics, machine learning

## Abstract

We present here COOBoostR, a computational method designed for the putative prediction of the tissue- or cell-of-origin of various cancer types. COOBoostR leverages regional somatic mutation density information and chromatin mark features to be applied to an extreme gradient boosting-based machine-learning algorithm. COOBoostR ranks chromatin marks from various tissue and cell types, which best explain the somatic mutation density landscape of any sample of interest. A specific tissue or cell type matching the chromatin mark feature with highest explanatory power is designated as a potential tissue- or cell-of-origin. Through integrating either ChIP-seq based chromatin data, along with regional somatic mutation density data derived from normal cells/tissue, precancerous lesions, and cancer types, we show that COOBoostR outperforms existing random forest-based methods in prediction speed, with comparable or better tissue or cell-of-origin prediction performance (prediction accuracy—normal cells/tissue: 76.99%, precancerous lesions: 95.65%, cancer cells: 89.39%). In addition, our results suggest a dynamic somatic mutation accumulation at the normal tissue or cell stage which could be intertwined with the changes in open chromatin marks and enhancer sites. These results further represent chromatin marks shaping the somatic mutation landscape at the early stage of mutation accumulation, possibly even before the initiation of precancerous lesions or neoplasia.

## 1. Introduction

Recent advances in DNA sequencing technologies have led to the development of reliable and cost-effective whole-genome sequencing methods and relevant analysis pipelines, which have been applied to numerous cancer types [1,2,3,4,5,6,7,8,9], precancerous lesions [4], normal tissues, cells, and stem cells [10,11]. The somatic mutation landscape from these data revealed various genotypic alterations, ranging from driver and passenger somatic mutations for cancer initiation and progression [12,13,14,15], mutation signatures [16], clonal and subclonal evolutions [17], to the findings on novel structural variations including kategis [18,19], chromothripsis [20], and whole-genome doubling [21,22]. Although mechanisms on how these aberrations arise have not been fully defined, a number of statistical and machine-learning approaches have shown that the somatic point mutation landscapes of multiple cancer types and precancerous lesions correlated with the chromatin mark landscape [23,24]. Tissue-of-origin (TOO) or cell-of-origin (COO) predictions for different cancer types are possible by leveraging such information and beyond [25,26,27,28,29,30].

To this day, tools used for these predictions mainly utilize random forest algorithm, an ensemble learning method which uses the averaged result derived from a predefined number of decision trees for the prediction. Random forest-based TOO/COO prediction algorithms, however, still face limitations such as low prediction accuracy for some cancer types; this might be due to the sparse mutation density and the requirement of a strong server-level computing power for reasonable speed. Extreme gradient boosting (XGBoost), on the other hand, employs gradient weighing based on the prior prediction running and similarity score-based tree pruning, which in the end is expected to be resistant to the unbalanced or sparse data, with improved running speed. Here we describe a novel XGBoost machine learning-based tool called COOBoostR, which displays a notable improvement in the prediction accuracy when there is low mutation density. It also offers the advantage of higher analysis speed, in addition to a minimal requirement of computing power. To assess the validity and accuracy of the algorithm, on the aspect of tissue or cell-of-origin predictions, we applied COOBoostR to the somatic point mutation landscape data from 636 tumors, 23 precancerous lesions, 35 normal stem cells, and 14 normal tissue clones/samples.

## 2. Results

COOBoostR receives mutation and chromatin data as the input after quality control (Figure 1a) (Online Methods). We then equally divide the genomic regions into 1-megabase and calculate the values of these data in each region, as has been reported previously [23]. Among the ensemble models, COOBoostR applies the XGBoost methodology to predict the TOO and/or COO. While the random forest-based TOO and COO prediction algorithms use a bagging model which performs parallel training, the boosting model of COOBoostR evaluates feature weights from sequential training. This training process improves the accuracy of the COO algorithm by strengthening models with difficult prediction. The output of COOBoostR is a ranked list of chromatin marks, where the higher rank represents higher similarity with regional somatic mutation density. Eventually, the tissue or cell type corresponding to these chromatin marks is assigned as potential TOO or COO.

To estimate the accuracy of COOBoostR algorithm, we conducted TOO prediction for melanoma, multiple myeloma, and glioblastoma samples (Figure 1b) [2,5,6]. For these samples, aggregate-sample level and individual sample level predictions using random forest-based algorithm were previously reported (prediction accuracy: 69.56% to 100%) [23]. Our analyses revealed that for melanoma samples, COOBoostR algorithm resulted in 64% accuracy for assigning correct TOO. In the case of multiple myeloma and glioblastoma samples, COOBoostR predicted correct TOOs with relatively higher accuracy (86.96% and 95.12%, respectively). To further assess speed and accuracy of COOBoostR, we compared the results from COOBoostR with a previously established random forest regression-based method [23,24] for the prediction of tissue-of-origin for colorectal cancer, esophageal adenocarcinoma (EAC), and liver cancer. These three cancer types were previously known to display higher variance explained score, compared to the other cancer types subjected to the random forest-based algorithm [23]. We were, at first, interested in evaluating the accuracy and reproducibility of the results after the repetitive running of COOBoostR. Our results displayed 100% reproducibility in 92.71% of tested samples (samples showing either 0% or 100% accuracy after 100 rounds of repetitions) (Appendix A). We also estimated trends in TOO accuracy according to the sample mutation density per each cancer type. Among the three cancer types, five out of six colorectal cancer samples showed 100% reproducibility when the mutation density per 1-megabase of the sample was greater than 4.7, whereas the EAC and liver cancer samples exhibited 100% reproducibility of the prediction results for 100% and 95.31% of the samples, regardless of the sample mutation density (Appendix A). Subsequently, we conducted an accuracy comparison between the COOBoostR and the random forest-based algorithm on these samples, with mutation density per 1-megabase greater than 4.7. For this, we measured the proportion of individual samples predicted as the correct tissue-of-origin with 100% reproducibility (Figure 1c). In liver cancer, both algorithms performed similarly well and predicted the correct tissue-of-origin with ~95% accuracy, whereas in colorectal and esophageal cancer, the COOBoostR algorithm predicted the tissue-of-origin in 1.25 and 1.05 times more samples compared to the random forest algorithm, respectively. In addition, we measured the algorithm running speed after applying COOBoostR or the random forest-based algorithm to the three cancer types. We found that COOBoostR completed the tissue-of-origin prediction process ~389.94 times faster than the random forest-based algorithm (Table 1). Taken together, we show that COOBoostR has reproducible results with better speed and comparable accuracy when compared to the existing random forest-based algorithm.

Next, we tested COOBoostR to evaluate whether the regional somatic mutation data of normal tissues or cells can be best explained by the epigenome of matching normal tissues or cell types. The first dataset we used for answering such a question was the whole-genome somatic mutation data from a normal liver [11]. To examine the extent of the TOO matching for a normal liver, COOBoostR was conducted in each individual sample from the dataset. Our results showed that all 14 individual samples, with mutation density ranging from 6.6 to 67.8, were accurately matched as liver TOO (Figure 2a). We next performed COOBoostR on individual clone-level data to see if the matching pattern could be replicated at the clonal level. Since the mutation density for the clone-level in this dataset is very low (0.80 per megabase on average per each clone), this helped us to test the performance of COOBoostR with sparse datasets, which was one of the intentions for creating XGBoost-based methods [31]. For this, we matched the TOO of normal liver samples at clone-level by selecting 86 clones harboring mutation density between 0.25 to 1.5 per 1-megabase, out of 491 clones extracted from 14 donors. Our results show a prediction accuracy of 50% when the somatic mutation density for each clone was ~1 per 1-megabase, and 80% if the mutation density was 1.25 or higher (Figure 2b). These results from normal liver tissues indicate that the extent of TOO matching accuracy could depend on the mutation density of the input data.

To further test the COOBoostR-based matching prediction efficiency and to corroborate our results from the analysis using liver normal tissues and clones, we subsequently ran COOBoostR on normal adult stem cell samples which contain lower amounts of mutations (0.09~1.38 per 1-megabase window) compared to the tumor tissues [10]. Previously, it has been reported that any cancer type with low mutation density displays relatively lower TOO predictive accuracy when using the random forest-based algorithm [23,24]; we anticipated that the robustness of the COOBoostR algorithm would at least partly compensate for such weakness. In the case of colon adult stem cells, we divided the samples into two age-based subgroups (young (age 9 to 15) and old (age 53 to 66)) to consider the predefined age stratifications inside the cohort (Figure 2c). COOBoostR based TOO matching results demonstrated an accuracy of 40% for the young age subgroup samples, and 75% prediction accuracy was obtained for old age subgroup samples. In contrast, random forest-based TOO predictions resulted in 0% accuracy for the young age subgroup and 62.5% for the old age subgroup, demonstrating comparative advantage of COOBoostR over the random forest-based algorithm. This trend was consistent in the case of small intestine adult stem cells (Figure 2d). Again, age-based subgroups were assigned (young (age 3 to 8), mid (age 44 to 45), and old (age 70 to 87)). For the old age subgroup, both algorithms predicted correct TOO with ~80% accuracy. However, COOBoostR-based TOO predictions for young and mid-age subgroups exhibited better performance compared to the random forest-based algorithm. While COOBoostR predicted correct TOO for 33.33% of the young age subgroup, and 100% of the mid-age subgroups, the random forest-based algorithm had an accuracy rate of 0% in the young age subgroup and 33.33% in the mid-age subgroup. Beyond an age-based subgrouping, we also checked whether there were any differences in TOO accuracy with respect to the mutation density per sample. For this, we arranged colon and small intestine adult stem cell samples according to the order of mutation density and assessed whether there was a trend. In the case of COOBoostR TOO prediction, accurate prediction was observed when the mutation density for colon adult stem cells was greater than 0.36, and the mutation density for small intestine adult stem cells was greater than 0.23 (Figure 2e). Conversely, in the case of random forest-based TOO prediction, we observed that a relatively high mutation density is required for consistent accurate prediction (colon adult stem cell mutation density greater than 0.85, small intestine adult stem cell mutation density greater than 0.61) (Figure 2f). In the case of liver adult stem cells, none of the samples were predicted as the expected TOO (liver tissue), which is in line with the previous result based on the random forest-based algorithm [26].

In addition to examining COOBoostR accuracy on the three cancer types (colorectal, EAC, and liver cancer) and normal tissue or stem cell types, we also wondered if the prediction accuracy of COOBoostR harbors a comparative advantage to the random forest-based method for hepatoblastoma [9], a pediatric neoplasm with the lowest somatic mutation density. Samples including mature (*n* = 18) and immature (*n* = 15) hepatoblastoma types, subjected to COOBoostR, had somatic mutation density of 0.133 per 1-megabase on average. When we set the liver tissue as a matching TOO for hepatoblastoma, the TOO prediction accuracy was 55.56% (5/9 samples) when the somatic mutation density was equal or higher than 0.2 per 1-megabase; whereas, the accuracy was 22.22% (2/9 samples) for the same mutation density window when utilizing a random forest-based algorithm (Appendix A). Collectively, COOBoostR displayed improved prediction accuracies for the samples harboring lower mutation density compared to the random forest-based algorithm, albeit still showing mutation density dependent differences in accuracy performance. Additionally, our results are still in line with the previous finding that the TOO prediction accuracy and the mutation density are interrelated, which might be still intrinsic to the tree-based machine learning algorithm.

Recently, it has been suggested that the Barrett’s metaplasia (BM), a representative precancerous lesion associated with tissue metaplasia, and the EACs primarily originate from gastric cells at the level of mouse lineage tracing models [32], and primary human samples [28,33]. To confirm whether COOBoostR prediction results align with the former outcomes, we employed Fixed-Tissue Chromatin Immunoprecipitation Sequencing (Fit-seq) data from four different tissues (BM, Squamous, Ileum, and Gastric antrum) (Online Methods), which has been previously utilized for the random forest-based TOO prediction [28]. As a first step, we tested the COOBoostR algorithm on BM samples and individual matching, paired EACs (*n* = 23). As a result, 22 out of 23 BM and EAC samples were predicted as gastric TOO, which is consistent with the gastric TOO predominancy observed by the random forest-based algorithm (Figure 3a). Subsequently, we examined whether the prediction would change when using 1-megabase region subsets containing certain tissue-specific enhancers (Online Methods). COOBoostR predicted gastric TOO for 12 out of 23 BM samples and 16 out of 23 EAC samples when using 1-megabase regions containing gastric tissue-specific enhancers; whereas, the proportions were relatively unchanged when using 1-megabase region subsets containing squamous tissue-specific enhancers (22 out of 23 samples for BM, 23 out of 23 samples for EACs) (Figure 3a). To examine whether these results are replicated in other sample sets, we assessed the TOO prediction accuracy for the samples from two other EAC studies [3,8]. As predicted, a total of 387 out of 409 EACs [3] and 9 out of 9 EACs [8] were predicted as gastric TOO (Figure 3b). In addition, the samples predicted as gastric TOO were decreased (296 out of 409 samples [3], 4 out of 9 samples [8]) when using 1-megabase regions containing gastric tissue-specific enhancers, whereas 390 out of 409 samples [3] and 9 out of 9 samples [8] were predicted as gastric TOO when using 1-megabase regions containing squamous tissue-specific enhancers. These results imply that the changes in chromatin marks, at or in the vicinity of the tissue-specific enhancer regions, are likely to affect the somatic mutation density profile at those regions.

To assess any potential bias in TOO prediction results due to the differences in the number of 1-megabase regions, we conducted COOBoostR using 673 chromatin marks on seven different cancer types, and BM at aggregate sample mutation level after randomly selecting different numbers of 1-megabase region subsets (ranging from 500 to 1000, Online Methods). Cancer type-dependent differences in TOO accuracies were observed only at 500 region subsets, but the prediction accuracy uniformly exceeded 97% when using 700 region subsets or more (Figure 3c). Additionally, the prediction accuracies of EACs [8] were the most consistent, reaching 100% regardless of the number of region subsets. To assess if the TOO accuracy levels are still reproduced when using a different chromatin dataset, we moved onto measuring COOBoostR prediction accuracies for EACs [8] with different region subsets utilizing the Fit-seq data. For this, we applied different region subsets according to the proportion of different enhancer containing regions (ranging from 0% to 100%, Online Methods). Results from this analysis demonstrated that the TOO prediction accuracy, using 500 or 1000 region subsets, was over 92% regardless of the enhancer containing region proportions, and the enhancer originating tissue types (Appendix A). However, the gastric TOO prediction accuracies using 100 region subsets mostly decreased, as the proportion of the gastric enhancer containing regions become higher; whereas, the prediction accuracy using 100 region subsets with different squamous enhancer containing region proportions was consistently high (over 88%) (Figure 3d). These results provide evidence of minimal bias towards the number of 1-megabase regions on COOBoostR prediction, and reinforce the role of tissue specific chromatin mark profiles on the somatic mutation accumulation during the course of precancerous lesions and cancer development.

## 3. Discussion

COOBoostR is based on the extreme gradient boosting method (XGBoost) [31], which was originally designed to cope with the issues raised by tree-based learning approaches (relatively low throughput with high consumption of computing power, poor prediction accuracies when using sparse data matrix, etc.). Based on the gradient weighing and tree pruning process embedded in the algorithm, we utilized the XGBoost methodology to select and rank tissue or cell-level chromatin mark features with respect to the relationship with somatic point mutation density of normal tissue/pre-cancerous lesions/tumor samples with improved speed and computing power compared to the existing random-forest based algorithm. Leveraging this machine learning based feature ranking, TOO/COO predictions on samples harboring diverse mutation densities were performed by incorporating different chromatin mark input datasets.

Although there are several publications which describe strong correlative measurements between TOO/COO chromatin marks and the somatic mutation landscape of tumors or precancerous lesions [23,24,26,29,34], no pre-existing reports tackled the question on whether the regional somatic mutation density profiles of normal tissues or cells indeed best explained by the matching TOO or COO mark profiles. We postulated that the accurate matching of TOO/COO for normal tissues or cell types, through utilizing our algorithm with somatic mutation inputs, would not only confirm that COOBoostR is working as designed, but also support the argument on the role of chromatin marks shaping the somatic mutation landscape; thus, such a concept could be utilized for predicting TOO/COO for various cancer types. Albeit revealing limitations, possibly due to the very low mutation density for some of the samples, COOBoostR showed that the regional somatic mutation data of normal tissues or cells can be best explained by the epigenome of matching normal tissues or cell types, which is more apparent when compared to the random forest-based algorithm for the samples harboring lower mutation density. In line with this, a similar phenomenon was observed for the hepatoblastoma samples, which showed the lowest mutational density among human cancers [9]. It would be worth investigating to see if this pattern applies to other normal tissues, cell types, and pediatric tumors.

In this study, we examined whether COOBoostR prediction results per individual sample would change when utilizing 1-megabase region subsets containing tissue-specific enhancers. This analysis was based on the hypothesis that the chromatin marks at or proximal to the TOO-specific enhancer regions would change during the progression to BM, mainly by the loss of such enhancers, and somatic mutation accumulation patterns inside those regions would change dynamically during the progression to BM. Thus, they would correlate less with the original TOO chromatin marks, and ultimately affect COOBoostR accuracy. In line with our hypothesis, the marked decreases in gastric TOO-predicted samples for BM and EAC were observed when using 1-megabase regions containing gastric tissue-specific enhancers, whereas the proportions of gastric TOO-predicted samples were conserved when using 1-megabase region subsets containing squamous tissue-specific enhancers. This phenomenon was recapitulated for the samples from two other studies, by demonstrating reduced TOO prediction accuracy when using 1-megabase regions containing gastric tissue-specific enhancers. This phenomenon was more evident when the number of region subsets became lower. Although the window size of the enhancer regions is far less than 1-megabase, our results do emphasize the changes in tissue specific chromatin marks during precancerous lesions/malignant tumor progression, at least in the case of BM and EAC, are sufficient to be captured as COOBoostR prediction accuracy changes. 

One of the areas which our current study did not tackle was running COOBoostR after stratifying mutation context patterns or mutation signatures. Since there is possibility of specific mutation context or signatures differing in somatic mutation accumulation patterns, and their relationship with the chromatin mark features, prediction accuracy of the TOO/COO algorithm can also be affected. Additionally, the prediction accuracy differences among cancer types shown in our work, and this might be intertwined with the cancer type specific mutation contexts and the prominence of mutation signatures, which would require future research to examine such effects.

In conclusion, COOBoostR showed at least comparable or better TOO/COO prediction speed and accuracy, compared to the random forest-based algorithm, when utilizing somatic mutation data from diverse types of samples ranging from tumors to precancerous lesions, and normal tissues. Additionally, our results provide several lines of evidence for dynamic somatic mutation accumulation at the normal tissue or cell stage, which could be intertwined with the changes in open chromatin marks and enhancer sites. Recent publications [35,36] start to shed light on the possible role of cancer cell-of-origin and tumor origin giving rise to different cancer types, and providing potential diagnosis methods for the cancer of unknown primary (CUP). In line with this, COOBoostR would facilitate cancer COO investigations and predictions in humans, which would be critical to early cancer diagnosis and selection of treatment options.

## 4. Online Methods

### 4.1. Somatic Mutation Data Derived from Whole-Genome Sequencing

We calculated regional somatic mutation density for 708 individual cancer genomes derived from several cancer types, precancerous lesions, normal stem cells, and normal tissues and cells. The International Cancer Genome Consortium (ICGC), which include Pan-Cancer Analysis of Whole genomes (PCAWG), Accelerating Research in Genomic Oncology (ARGO), and European Genome-Phenome Archive (EGA), have granted permission to use 64 liver cancer genomes (LIV) [1], 41 glioblastoma genomes (GBM) [2], 413 esophageal adenocarcinoma genomes (EAC, Frankell. et al.) [3], and 23 pairs of Barrett’s metaplasia (BM) matching with esophageal adenocarcinoma genomes (EAC, Ross-Innes. et al.) [4]. In our study, Barrett’s metaplasia genomes were employed as a representative case of precancerous lesions. In the case of 413 EACs deposited in ICGC ARGO, four samples (DO234285, DO234363, DO234413, DO234462) with hypermutations were excluded from TOO/COO predictions and the genomic coordinates of the variants for each sample were converted from Genome Reference Consortium Human Build 38 (GRCh38) to Genome Reference Consortium Human Build 37 (GRCh37) using CrossMap [37]. From The database of Genotypes and Phenotypes (dbGaP), we have been granted authority for data use of 25 melanoma genomes (MEL) [5], 23 multiple myeloma genomes (MM) [6], 9 colorectal cancer genomes (CRC) [7], and 9 esophageal adenocarcinoma genomes (EAC, Dulak. et al.) [8]. We have been also granted access to published 33 hepatoblastoma genomes from Hiroshima University and RIKEN [9]. For normal stem cells, we gathered tissue-specific somatic mutation accumulation in adult stem cells from publicly available datasets [10], which included 21 colon stem cell genomes and 14 intestine stem cell genomes. As a representative case of normal tissue, we not only extracted 14 normal liver genomes, but also 491 normal liver clone genomes within samples from the published data source [11]. Since the mutation rate of normal liver clone genomes per 1-megabase window were distributed from 0.25 to 1.5, we divided each interval into 0.25 mutation rate units, and randomly selected a total of 86 clonal samples from each interval. In order to measure the regional mutation density for each sample, autosomes were split into 1-megabase regions excluding areas related to centromeres, telomeres, and low quality unique mappable base pairs. After that, we aggregated the frequency of variations in each 1-megabase region and established somatic mutation profile for individual samples. This mutation counting process was carried out using BEDOPS [38], and based on the Genome Reference Consortium Human Build 37 (GRCh37).

### 4.2. Chromatin Data Derived from Chromatin Immunoprecipitation Sequencing (ChIP-Seq)

A total of 673 ChIP-seq data, which include human primary tissues and cell lines, were extracted from ENCODE [39], IHEC [40], and the NIH Roadmap Epigenomics Consortium (release 9) [41]. In our study, these ChIP-seq features were classified as a total of 132 tissue or cell types according to the original source [29]. Moreover, the ChIP-seq data utilized in this study contain two kinds of repressive histone modifications (H3K27me3 and H3K9me3) and five kinds of active histone modifications (H3K27ac, H3K36me3, H3K4me1, H3K4me3, and H3K9ac). Consistent with the mutation profile counting, we calculated ChIP-seq reads count in each 1-megabase region based on the human genome version GRCh37 (hg19).

### 4.3. COOBoostR: XGBoost-Based Feature Selection Algorithm for TOO/COO Prediction

Identification of putative TOO/COO for cancer types can be crucial for decoding the responsible tissue or cell types that should be subjected to monitoring-based cancer prevention, and understanding the somatic mutation accumulation mechanisms intertwined with the cancer development. Pre-existing methods for TOO/COO identifications include lineage-tracing mouse models and organoids, which are either heavily depending on particular genetically engineered mouse models or are time-consuming with extremely low throughput. Bioinformatics and a machine learning based approach to predict TOO/COO for human tumor samples are unique in a sense that they are the sole practical method for estimating the TOO/COO directly for human samples with a reasonable throughput. 

XGBoost is an extendible and state-of-the-art algorithm of gradient boosting machines which has proven to push the limits of computing power for boosted trees algorithms [31]. It was developed for the sole purpose of model performance and computational speed. Boosting is an ensemble technique in which new models are added to adjust for existing model errors. Model is recursively added until the error is no longer small and is adjusted according to the hyperparameter. Furthermore, gradient boosting is an algorithm that combines the previous model with the new model that predicts the residuals of the previous model to make the final prediction, which then updates the weights of the model. To minimize the loss when adding new models, gradient descent algorithm is utilized. Additionally, the performance was significantly improved by using multiple cores of a CPU and reducing the lookup times of individual trees created in the XGBoost.

COOBoostR was created to achieve higher accuracy and speed compared to the current state-of-the-art random forest-based algorithm, by taking advantage of the XGBoost algorithm. The input of COOBoostR is a matrix containing regional somatic mutation density for each 1-megabase window (2128 rows) of the human genome, whereas the output provides top 20-ranked chromatin marks derived from responsible tissue or cell types which show high correlations to the regional somatic mutation densities. For this algorithm to work accordingly, we applied backward elimination manners to seek a minimal set of predictors for each genome. Additionally, we trained the COOBoostR model with 10-fold cross-validation on the complete set of variables, and determined the importance of all the variables in the model. We then ranked the predictors according to their importance and determined the top 20 variables. The most important variable (top 1) is obtained through the training 20 models and the backward elimination method. The last predictor variable represents the most similar landscape regarding the 1-megabase level regional mutation density. 

Tuning COOBoostR can involve many hyperparameters including “n estimators”, which determine the epoch of the model, “learning rate”, gamma”, “max depth”, etc. To customize the model, hyperparameter tuning is possible. For example, several learning rate variable options (0.05, 0.1, 0.3, 0.4, 0.5, 0.6, 0.7, 0.8, 1.0) are available, of which 0.5 was selected as a default value. In addition, a default value of 20 was set for the n estimators variable, and a gamma of 0 (0, 0.5, 1, and 2 can be options) was set as a default. Finally, max depth of 6 was used as the default value among 1, 3, 6, and 8.

### 4.4. Random Forest-Based TOO/COO Prediction

Our TOO/COO predictive analysis based on random forest regression was performed by reflecting and modifying previous research [23,24,26]. Once the training sets of each tree were constructed, the mean squared errors were then measured from out-of-bag data to determine the importance of each variable. In each tree, the values of these variables were randomly permutated and estimated. The raw importance value of variable m was calculated by subtracting the mean squared error between the untouched out-of-bag data and the variable-m-permuted data. Consequently, the importance ranking of each variable was estimated from the average score of the variable m in the entire tree. To predict regional mutation density for each sample, we generated 1000 random forest trees based on 673 epigenetic features. In addition, the TOO/COO for each sample was predicted from the tissue/cell type of top 1 epigenetics marker, identified by employing the greedy backward elimination method. The random forest models at each stage were repeatedly tested 1000 times. We employed these random forest-based TOO/COO prediction algorithms to compare the speed and accuracy with the COOBoostR algorithm. For the speed comparison between the two algorithms, the genomes of liver cancer, esophageal adenocarcinoma (Ross-Innes. et al.), and colorectal cancer were employed. In comparing the prediction accuracy of the two algorithms, not only these cancer type samples, but also normal adult stem cell genomes were employed.

### 4.5. COOBoostR and Region Subset-Based Analysis

Aggregated regional somatic mutation density for 5 cancer types: esophageal adenocarcinoma from Ross-Innes. et al. (EAC), esophageal adenocarcinoma from Dulak. et al. (EAC), colorectal cancer (CRC), multiple myeloma (MM), and Glioblastoma (GBM)) and 1 pre-cancerous lesions (Barrett’s Esophagus (BE)) were subjected to the analyses. For generation of aggregated regional somatic mutation density matrix, the number of somatic mutations per 1-megabase region was added up for all of the samples corresponding to each cancer type. Additionally, individual sample-level regional somatic mutation density for EAC (5 samples) cancer type was utilized for the analyses. These individual samples were selected based on their COOBoostR prediction results using entire regions matching to the TOO/COO prediction performed by using aggregated regional somatic mutation density of the corresponding cancer type.

For aggregated regional somatic mutation density data, COOBoostR was performed on 6 different region subset cases (500, 600, 700, 800, 900, and 1000 regions), and the accuracy (defined by predicted TOO/COO when using all of the 2128 regions) calculated by 100 random sampling iterations was derived for each case. For individual sample-level regional somatic mutation density data, subset cases consisted of a total of 21 different number of regions (50, 100, 150, 200, 250, 300, 350, 400, 450, 500, 550, 600, 700, 800, 900, 1000, 1200, 1400, 1600, 1800, and 2000 regions), and the accuracy measurements were derived by 20 times of random region subsampling.

### 4.6. Tissue Specific Enhancer Containing Region Subset Based TOO Predictions

For the region selecting-based TOO prediction of BM and EACs containing tissue specific enhancers, Fixed-Tissue Chromatin Immunoprecipitation Sequencing (Fit-seq) data derived from four different tissues (Gastric, Barrett’s, Squamous, Ileum) were employed [28]. We first selected the regions containing tissue specific enhancer regions (gastric or squamous) from a total of 2128 regions. Then, among the four types of Fit-seq markers, samples predicted as a gastric tissue mark were considered as the accurate TOO predicted samples for both BM and EACs. In Figure 3a,b, we measured the accuracy of TOO prediction for BM and EACs at the individual sample level. In Figure 3d and Appendix A, TOO accuracy was measured after subsetting regions based on the proportion of tissue specific enhancer containing regions (0, 25, 50, 75 to 100% for 5 EACs (covering the lowest to the highest somatic mutation density within the sample set). Specifically, a random selection of the enhancer containing regions is first performed, with respect to the predefined proportion of the enhancer containing regions, then the random selection of the other regions is conducted to fill the total number of the intended subset regions if there is any room left. A total of 100 rounds were executed per each sample and the fixed region number, and then the TOO prediction accuracy was calculated.

### 4.7. Code Availability

Source code for COOBoostR is available at https://github.com/SWJ9385/COOBoostR (accessed on 19 December 2022).

## Figures and Tables

**Figure 1 life-13-00071-f001:**
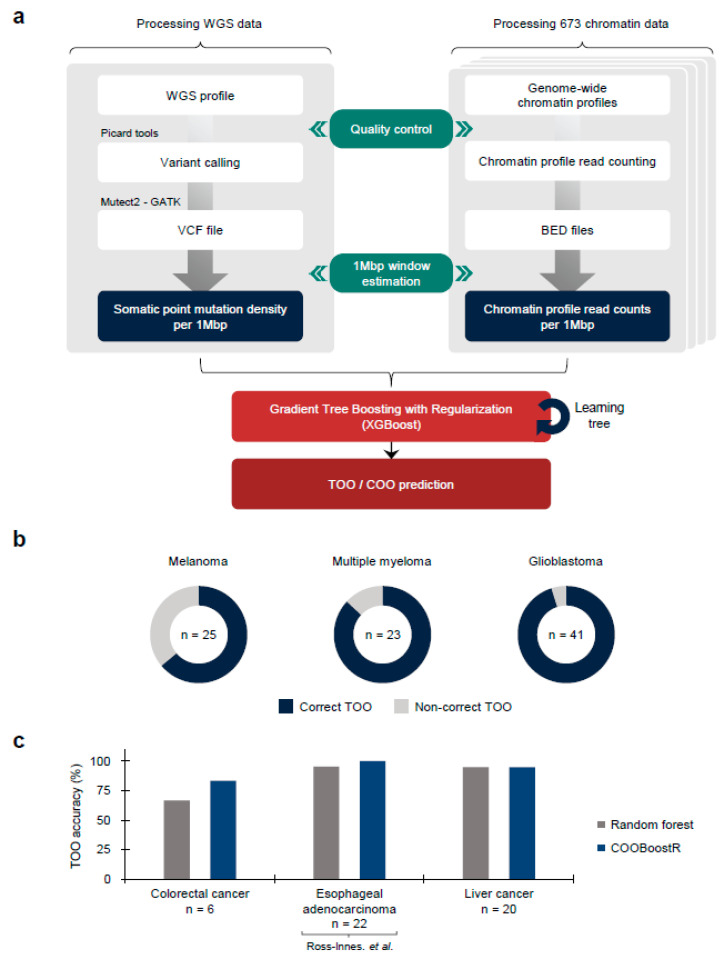
TOO/COO Prediction with COOBoostR. (**a**) COOBoostR algorithm flow diagram. A total of 708 WGS data and 673 chromatin data were subjected to the 1-megabase level preprocessing followed by XGBoost-mediated prediction stage. (**b**) TOO prediction accuracy for melanoma, multiple myeloma, and glioblastoma with COOBoostR. (**c**) TOO accuracy comparison between COOBoostR and random forest-based algorithm for colorectal cancer, esophageal adenocarcinoma [4], and liver cancer.

**Figure 2 life-13-00071-f002:**
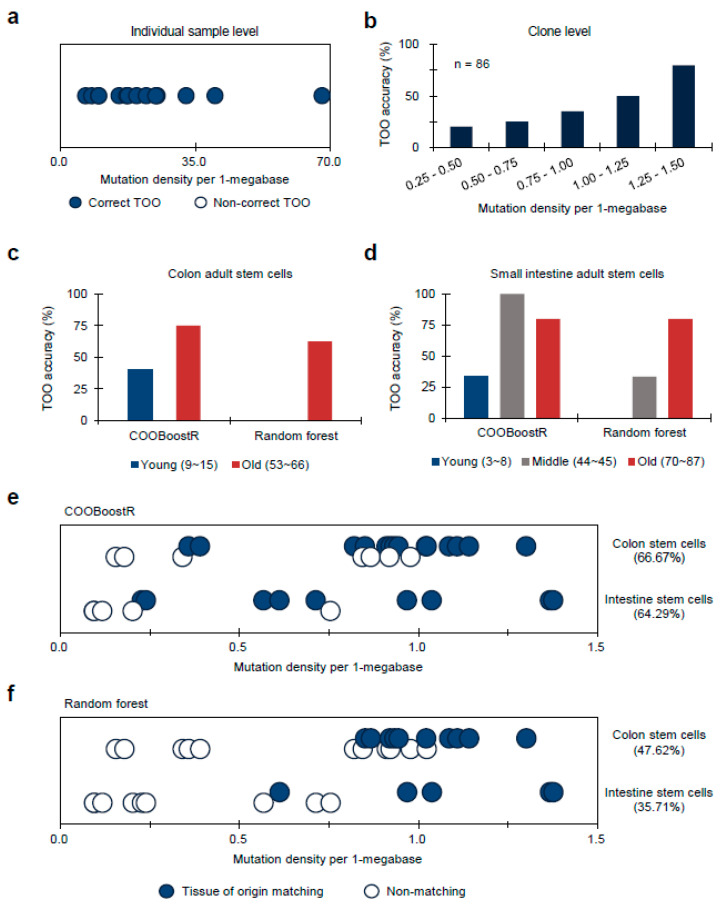
TOO prediction accuracy for normal stem cell organoids or tissue samples harboring low mutation density (average mutation density for stem cell organoids: 0.73, liver tissue clones: 0.63). (**a**) TOO prediction accuracy for normal liver at individual sample level. Samples are aligned in order of mutation density magnitude per 1-megabase window after classifying the correctness of TOO prediction. Dots were jittered to dissect out the blue and white dots. (**b**) TOO prediction accuracy for normal liver at clone level. Histogram showing TOO prediction accuracy of liver clones with respect to the mutation density groups. (**c**) TOO accuracy comparison for colon adult stem cells according to age-based subgrouping between COOBoostR and Random forest-based algorithm. (**d**) TOO accuracy comparison for small intestine adult stem cells according to age-based subgrouping between COOBoostR and Random forest-based algorithm. (**e,f**) TOO prediction accuracy for colon and intestine stem cell organoids at an individual sample level using COOBoostR (**e**) or Random forest-based algorithm (**f**). Samples matching predicted TOO are marked with solid circles, and samples that did not match are marked with empty circles. Dots were jittered to dissect out the blue and white dots. Samples are aligned in order of mutation density magnitude per 1-megabase window.

**Figure 3 life-13-00071-f003:**
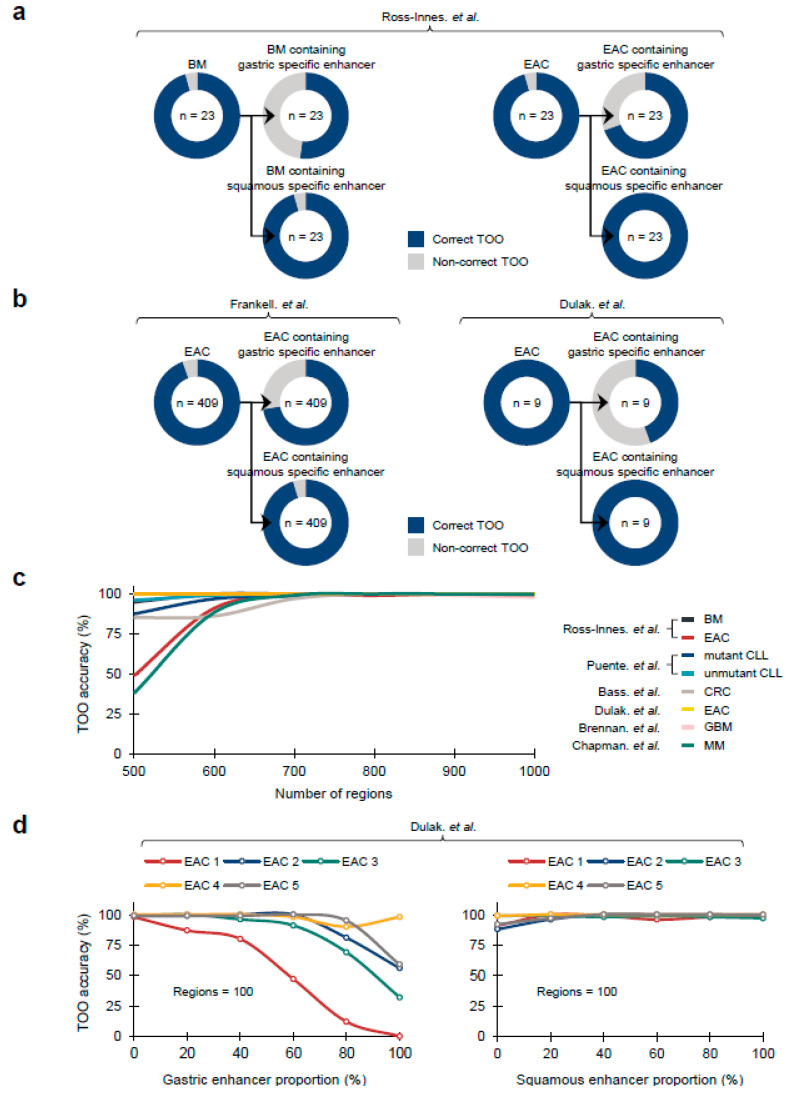
Region selection based TOO accuracy measurement for COOBoostR. (**a,b**) TOO prediction accuracy for BM and EAC derived from different study sources. In each case, TOO prediction accuracy was again measured by utilizing 1-megabase regions containing gastric specific enhancers or squamous specific enhancers. BM: Barrett’s metaplasia; EAC: Esophageal adenocarcinoma. (**c**) Region selection based COOBoostR accuracy measurement ranging from 500 to 1000 regions for 8 sample types (7 cancer types, 1 precancerous lesion) at aggregated sample level. (**d**) Region selection analysis with respect to the portion of gastric/squamous specific enhancer containing regions for EACs at individual sample level. Enhancer inclusion ratio was varying from 0 to 100%, making up to 100 regions [2,3,4,6,7,8].

**Table 1 life-13-00071-t001:** Speed comparison between COOBoostR and random-forest algorithm.

Cancer Type	n	Inspection Type	Average Time (s)	Min Time (s)	Max Time (s)
Colorectal cancer	9	Random forest 1R	316,777	303,240	335,601
		COOBoostR 1R *	8	6	11
		COOBoostR 100R	761	577	1130
Esophageal adenocarcinoma Ross-Innes. et al. [4]	23	Random forest 1R	447,082	409,602	475,357
		COOBoostR 1R *	16	10	26
		COOBoostR 100R	1588	1047	2615
Liver cancer	64	Random forest 1R	456,407	370,398	720,583
		COOBoostR 1R *	7	4	15
		COOBoostR 100R	707	383	1484

* COOBoostR 1R values were estimated from COOBoostR 100R investigation.

## Data Availability

Not applicable.

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
