# Peer review of "COOBoostR: An Extreme Gradient Boosting-Based Tool for Robust Tissue or Cell-of-Origin Prediction of Tumors"

_life, 2022, doi:10.3390/life13010071_

Round 1

Reviewer 1 Report

This manuscript describes an application of gradient boosting to the problem of cell / tissue of origin prediction in cancer genomics. The authors release a corresponding software COOBoostR. They show that COOBoostR shows at least comparable and likely superior performance compared to Random Forest used in the original publication. I have no technical concerns. However, it is somewhat limiting that the authors have only attempted to use a new algorithm but did not investigate the problem at a deeper level. For example, they did not attempt a dynamical approach to the spatial scale and kept 1Mb scale for all tumors. They also did not attempt to stratify mutations by type or context dependency. To sum, the manuscript provides a methodological advance but could take an extra step.

Author Response

Comment 1: However, it is somewhat limiting that the authors have only attempted to use a new algorithm but did not investigate the problem at a deeper level. For example, they did not attempt a dynamical approach to the spatial scale and kept 1Mb scale for all tumors.

> Thank you for your comment on fixing the spatial scale to 1Mbp window and its rationale. We chose 1 Mbp window based on the following reason:

This 1Mbp window size was not arbitrary but was frequently used on related publications from three different groups. Polak et al (DOI: 10.1038/nature14221) used 1 Mbp window in their entire paper, including random forest algorithm based feature selection and other analyses. In the case of Schuster-Bockler et al (DOI: 10.1038/nature11273), they also used 1 Mbp window for their core analyses on assessing the correlation between cancer SNVs and different chromatin levels. In addition, this paper showed that the correlation level was lower for smaller windows (10kb and 100kb) comparing to larger windows (1 Mbp and 10 Mbp), which could be due to low median number of SNVs per window. Recently, Nguyen et al (DOI: 10.1038/s41467-022-31666-w) also developed a random forest tissue-of-origin classifier employing complex somatic driver and passenger mutations based on 1 Mbp window.

Comment 2: They also did not attempt to stratify mutations by type or context dependency. To sum, the manuscript provides a methodological advance but could take an extra step.

> Thank you for the suggestion. For this, we consider such research as one of the next research areas with the highest priority. We incorporated this comment in discussion section (page 10, line 320 to 327)

Reviewer 2 Report

Overall, it is an interesting task. The task can be further improved by:

a-      In abstract section, need to mention the accuracy of algorithm

b-      Need to improve abstract, more clearly mention the method.

c-      In introduction section, cite some relevant article.  

1)      Iqbal, Muhammad Shahid, et al. "Deep learning recognition of diseased and normal cell representation." Transactions on Emerging Telecommunications Technologies (2020): e4017.

Author Response

Comment 1: In abstract section, need to mention the accuracy of algorithm

> Thank you for the comments about the need to describe algorithm accuracy. A summary of accuracy description is written below and the average accuracy % numbers are included inside the abstract. (page 1, line 31 to 32)

Comment 2: Need to improve abstract, more clearly mention the method.

> Thank you for the comments about the need to describe the method more clearly. We applied your comment inside the abstract (page 1, line 26 to 27)

Comment 3: In introduction section, cite some relevant article.

> I appreciate you suggesting the pertinent article. We therefore incorporated this study into the introduction. (page 2, line 53)

Reviewer 3 Report

The manuscript entitled “COOBoostR: an extreme gradient boosting-based tool for robust tissue or cell-of-origin prediction of tumors” describes the novel computational tool for predicting the tissue- or cell-of-origin of different cancers based on the integration of data on regional somatic mutation density and the data on chromatin marks. The authors offer an alternative to the commonly used tools which employ random forest algorithms, aiming to overcome the main limitations of such algorithms. Furthermore, accuracy of the COOBoostR was tested by analyzing the mutation landscape data for a large number of cancers, precancerous lesions and normal tissue/cells with different mutation density. Therefore, the main contribution of this manuscript relies on the improvements in the accuracy and the speed of tissue- or cell-of-origin of cancer prediction tools.

The manuscript is well structured, the applied methodology was adequate for this type of study and the main results are clearly presented. Still, some clarifications and elaborations are needed:

-       In the results section, data on prediction accuracy of COOBoostR algorithm are presented for melanoma, multiple myeloma, and glioblastoma samples and it is stated that the results obtained by using random forest-based algorithm for tissue-of-origin prediction were previously reported. Still, the authors did not state the exact prediction accuracy for random forest-based algorithm in previous publication and did not compare their results with previous ones.

-       A discussion would benefit from further elaboration on the potential clinical utility of the prediction of the cell-of-origin and the tissue-of-origin of cancer.

Author Response

Comment 1: In the results section, data on prediction accuracy of COOBoostR algorithm are presented for melanoma, multiple myeloma, and glioblastoma samples and it is stated that the results obtained by using random forest-based algorithm for tissue-of-origin prediction were previously reported. Still, the authors did not state the exact prediction accuracy for random forest-based algorithm in previous publication and did not compare their results with previous ones.

> Thank you for comments about the need to describe the prediction accuracy for random forest-based algorithms. We incorporated your comment inside the relevant result section. (page 3, line 93)

Comment 2: A discussion would benefit from further elaboration on the potential clinical utility of the prediction of the cell-of-origin and the tissue-of-origin of cancer.

> Thank you for comments about the need to describe the potential clinical utility. We wrote additional sentence inside the discussion to incorporate your comment. (page 10, line 334 to 336)